# Enhancing Multi-view Graph Neural Network with Cross-view Confluent Message Passing

## ABSTRACT

With the growing diversity of data sources, multi-view learning methods have attracted considerable attention. Among these, by modeling the multi-view data as multi-view graphs, multi-view Graph Neural Networks (GNNs) have shown encouraging performance on various multi-view learning tasks. The message passing is the critical mechanism empowering GNNs with superior capacity to process complex graph data. However, most multi-view GNNs are designed on the well-established overall framework, overlooking the intrinsic challenges of the message passing on multi-view scenarios. To clarify this, we first revisit the message passing mechanism from a graph smoothing perspective, revealing the key to designing a multi-view message passing. Following the analysis, in this paper, we propose an enhanced GNN framework termed Confluent Graph Neural Networks (CGNN), with Cross-view Confulent Message Passing (CCMP) tailored for multi-view learning. Inspired by the optimization of an improved multi-view graph smoothing problem, CCMP contains three sub-modules that enable the interaction between graph structures and consistent representations, which makes it aware of consistency and complementarity information across views. Extensive experiments on four types of data including multi-modality data demonstrate that our proposed model exhibits superior effectiveness and robustness.

## CCS CONCEPTS

• **Computing methodologies** → **Semi-supervised learning settings**; **Neural networks**.

## KEYWORDS

Graph neural networks, Message passing, Multi-view learning, Representation learning

## 1 INTRODUCTION

Multimedia technology has constantly developed, allowing data to be collected in a variety of ways. For instance, text, images and social relationships are used to describe personal information of the same group of users. This type of data including multiple sources can be referred to as multi-view data that provides diverse perspectives of an entity and retains richer information than single-view data [5, 27]. For comprehensively utilizing features from multiple

**Unpublished working draft. Not for distribution.**

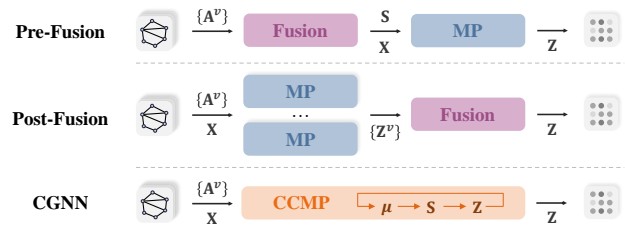

**Figure 1: Top:** Pre-fusion models integrate multi-view graphs into a whole graph for a consistent representation. **Middle:** Post-fusion approaches separately implement message passing on each graph followed by merging the obtained results. **Bottom:** Our CGNN achieves interactions between graph structures and consistent representations.

views to learn intrinsic representations, many traditional multi-view algorithms have been proposed in a variety of situations, including data mining [20, 39] and natural language processing [4, 23]. These methods have achieved excellent performance in various downstream tasks. However, the inherent or implicit connections between entities exist in real scenarios, and graphs are a data form capable of modeling such complex relationships. Benefiting from the powerful expressive capability of graphs, numerous traditional multi-view methods have been extended to graph learning and have achieved fulfillment [35, 44]. They attain a consistent graph structure via projecting graphs learned by multiple views onto a shared subspace or adopting the weighted fusion manner. Although these approaches successfully integrate the knowledge implied in graphs, they are limited in ego information and fail to benefit from the message passing.

The knowledge-fitting capability of neural networks enables complicated patterns and semantics in data to be captured. Graph Neural Networks (GNNs) [22], as powerful deep learning-based methods for propagating signals along graph-structured data, expand the convolution to non-Euclidean space and have become one of the hottest topics in numerous domains, including fraud detection [38, 49], computer vision [26, 53] and other areas [9, 33]. To leverage GNNs to mine more complex contacts and principles among multiple views, multi-view GNNs have emerged. For example, Li et al. [25] firstly combined the multi-view learning and GNNs to adaptively excavate relationships among views. Wu et al. [46] proposed an interpretable network to explore the multi-view data from the feature and topology spaces. Gong et al. [14] leveraged the self-paced way to add improved pseudo labels from other views to the target view to further optimize the model. Nonetheless, they learn the fixed graph based on the $k$-nearest neighbor algorithm or the distance between node pairs to perform the message passing, which treats vital and noisy edges as equal.

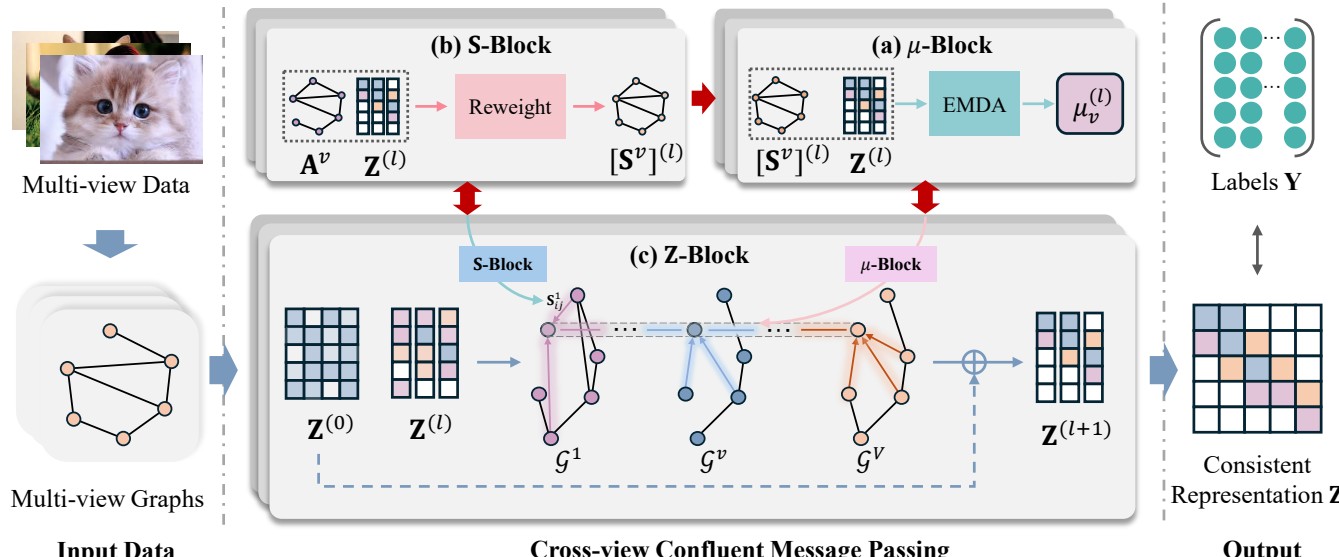

**Figure 2: The overview of the proposed CGNN. The network contains three sub-modules: (a) S-Block aims to learn edge-level coefficients. (b) μ-Block explores view-level coefficients. (c) Z-Block learns the consistent representation across views.**

As one of the critical steps of GNNs, the message passing mechanism performed on graph structures learns rich contextual semantics using relationships between nodes. Thus, most studies have been devoted to enhancing the graph structure by adjusting edges for discriminative node representations [24, 40]. GAT [37], as a classic method for graph enhancement, utilizes learned representations to assign different weights to node pairs, thereby filtering out abnormal connections. Following this line of thought, diverse variants [19, 47] are proposed to process complex multi-view graphs. Among these, pre-fusion and post-fusion mechanisms are two main pipelines for learning a consistent representation. The first one consolidates multi-view graphs first to capture an overall graph, and then it is used to accomplish the message passing for the consistent representation, as illustrated in Figure 1-Top. The second one realizes the message passing separately on each graph, and then integrates gained representations into consistent results, as shown in Figure 1-Middle. These approaches built upon well-established networks have exhibited superb performance by modeling multi-view data as multi-view graphs. However, they overlook interactions between graph structures and consistent representation, in which the latter can direct the augmentation and incorporation of the former for more discriminative representations.

To address the above challenges, we propose an effective multi-view GNN framework, named Confluent Graph Neural Networks (CGNN), with Cross-view Confluent Message Passing (CCMP). Concretely, we first reveal the key to devising a multi-view message passing by revisiting the message passing mechanism from a graph smoothing perspective. To enhance information aggregation from complex messages across various views, we introduce an explicit optimization objective that considers the interaction between multi-view graph structures and consistent representations. The layers in CCMP are transparently derived through alternating optimization of the objective, which includes three sub-modules dynamically

supervising from each other: S-Block exploring graph structures, μ-Block learning view-level coefficients, and Z-Block excavating consistent representations, as displayed in Figure 1-Bottom. These three sub-modules facilitate cross-view message flow at the node level. Specifically, each edge senses consistent information encoded in the learned representation and the importance of different views. Subsequently, the message passing is applied in the optimized graphs with consistent and complementary awareness to learn the final representation. The flowchart of CGNN is shown in Figure 2. Our contributions are summarized as

- We bridge multi-view message passing to the graph smoothing problem of GNNs and further reveal the key to designing multi-view message passing.
- We propose a node-level message passing induced by an improved multi-view graph smoothing problem and then integrate it into an efficient and robust model, termed, CGNN.
- Extensive experiments on four types of data indicate the superiority of the proposed framework. Moreover, CGNN works robustly on graphs with noisy edges.

## 2 RELATED WORK

### 2.1 Multi-view Leaning

For capturing the consistency and complementarity hidden in multi-view features to gain intrinsic semantics, many traditional algorithms have been proposed. For example, Xu et al. [50] leveraged relationships among views to recover the missing instances from a shared subspace. Cai et al. [3] designed a scaled simplex representation to achieve non-negative coefficients, and adopted the tensorized manner to explore the common and complementary information across views. Actually, connections between entities exist in real scenarios, and graphs, as a data form capable of describing such interactions, can aid in modeling complex data. Due

to the powerful expressive capability of graphs, graph learning and multi-view methods are combined to learn informative representations. For instance, Fang et al. [12] combined the adaptively integrated bipartite graphs with the learning of clustering structure to efficient clustering. Zhou et al. [56] constructed a graph filter based on multiple graph learning to explore the information from all views. These approaches can integrate the knowledge implied in graphs into objectives or constraints, thereby exploring the consistency and complementarity in multi-view graphs. In order to apply the advantages of neural networks to explore deep semantics among views, deep-based multi-view networks have been researched. Liu ey al. [28] introduced autoencoders to view-specific features and used contrastive learning to extract the common information among views. Jin et al. [21] constructed the sample-level alignment to mine the intra-view relationships and capture the inter-view semantics by prototype-to-prototype correspondence.

## 2.2 Message Passing Mechanism

To handle a large amount of graph-structured data present in the real world, GNNs-based paradigms have been extensively studied to perform the message passing along the graph structure for discriminative representations. However, noise and interference inherent in graphs could significantly impact the expressive power of GNNs, which is detrimental to downstream tasks. In light of this, many previous results have been devoted to improving graph structures and achieving notable accomplishments [11, 51]. These methods of modeling graph structures can be divided into three pipelines. The first is similarity-based approaches that utilize inner products or Gaussian kernel functions to compute the similarity between node representations, thereby assigning weights to edges. For example, Zhu et al. [58] computed the weight edges by fusing the topological and semantic knowledge from graphs to render the learned representations robust. The second approaches involve using neural networks to learn edge weights. For instance, Zhao et al. [55] designed neural edge predictors to enhance the intra-class links and separate the inter-class connections. The third modeling algorithms construct a learnable parameter matrix that is continually optimized during network optimization processes. For example, Ying et al. [52] utilized a mutual information-based loss to identify the important subgraphs. These models utilize optimized graph structures to aggregate neighbors for obtaining reliable representations. However, they lack interactions between graph structures and consistent representation.

## 3 PROPOSED METHOD

*Notations.* The major notations used throughout the paper are described below. Consider a multi-view graph $\mathcal{G} = \{\mathcal{G}^v\}_{v=1}^V$, where $V$ is the number of views and $\mathcal{G}^v$ is the $v$-th view data. For the $v$-th view $\mathcal{G}^v$, we have the adjacency matrix $\mathbf{A}^v \in \{0, 1\}^{n \times n}$. The graph Laplacian matrix is defined as $\mathbf{L}^v = \mathbf{D}^v - \mathbf{A}^v$, where $\mathbf{D}_{ii}^v = \sum_j \mathbf{A}_{ij}^v$ and $\mathbf{A}_{ij}^v = 1$ denotes the existence of edge $e_{ij} \in \mathcal{E}$ linking node $v_i$ and $v_j$ in the $v$-th view. $\mathbf{X} \in \mathbb{R}^{n \times m}$ is the feature matrix containing the node information, where each node is associated with an $m$-dimensional feature vector.

## 3.1 Understanding Message Passing via Graph Smoothing

**Insights into Message Passing:** We first review the key component of GNNs, that is, the message passing mechanism. Take the classical GCN as an example, the message passing process can be typically formed as a matrix-form graph convolution:

$$\mathbf{Z}^{(l+1)} = \hat{\mathbf{A}}\mathbf{Z}^{(l)}, \tag{1}$$

where $\hat{\mathbf{A}} = \tilde{\mathbf{D}}^{-\frac{1}{2}}\tilde{\mathbf{A}}\tilde{\mathbf{D}}^{-\frac{1}{2}}$, $\tilde{\mathbf{A}} = \mathbf{A} + \mathbf{I}$, and $\tilde{\mathbf{D}}$ is the degree matrix of $\tilde{\mathbf{A}}$. Some previous work has revealed that Eq. (1) equals to iteratively solve the following graph smoothing regularization:

$$\min_{\mathbf{Z}} \ \text{Tr}(\mathbf{Z}^\top \hat{\mathbf{L}}\mathbf{Z}), \tag{2}$$

where the normalized Laplacian $\hat{\mathbf{L}} = \mathbf{I} - \hat{\mathbf{A}}$ is adopted. Actually, Problem (2) can be rewritten in a node-form:

$$\min_{\mathbf{Z}} \ \frac{1}{2}\sum_{i,j} \mathbf{A}_{ij}\|\bar{\mathbf{z}}_i - \bar{\mathbf{z}}_j\|_2^2, \tag{3}$$

where $\bar{\mathbf{z}}_i = \mathbf{z}_i/\sqrt{d_i}$ and $d_i = \tilde{\mathbf{D}}_{ii}$. It imposes a graph smoothing regularization that makes the representations of any two connected nodes more similar in the learned low-dimensional space. This node-form graph smoothing regularization essentially explains the message passing operation. More specifically, for the $i$-th node, the message passing is performed by

$$\mathbf{z}_i^{(l+1)} = \frac{1}{d_i}\mathbf{z}_i^{(l)} + \sum_{j \neq i} \mathbf{A}_{ij}\frac{1}{\sqrt{d_i d_j}}\mathbf{z}_j^{(l)}, \tag{4}$$

which collects the message from neighbors of the node $i$ to update its representation. Problem (3) provides an insightful understanding of the message passing mechanism, and we further analyze it in the multi-view scenario.

**Towards Multi-view Message Passing:** Although the multi-view graph typically contains diverse types of relations, it potentially possesses cross-view consistent and complementary information, as these relations share the same node set. Therefore, multi-view GNNs aim to integrate information from these complex relations to learn a consistent representation. For example, from the perspective of graph smoothing, a typical multi-view GNN corresponds to the following graph smoothing problem:

$$\min_{\mathbf{Z}} \ \frac{1}{2}\sum_v \sum_{i,j} \mathbf{A}_{ij}^v\|\bar{\mathbf{z}}_i - \bar{\mathbf{z}}_j\|_2^2. \tag{5}$$

However, the interaction between consistent representation and relations is not well-considered, such that the consistency and complementarity encoded in representation $\mathbf{Z}$ are learned on fixed graph structures. In other words, the edges are not aware of cross-view consistency. To tackle this, we design a better multi-view graph smoothing:

$$\min_{\mathbf{Z}} \ \frac{1}{2}\sum_v \sum_{i,j} \omega_{ij}^v\mathbf{A}_{ij}^v\|\bar{\mathbf{z}}_i - \bar{\mathbf{z}}_j\|_2^2, \tag{6}$$

where $\omega_{ij}^v$ is an edge-level coefficient calculated on $\mathbf{z}_i$ and $\mathbf{z}_j$, which measures the importance of edge $e_{ij}$ to the consistent representation $\mathbf{Z}$. Although this formulation weights each edge for every view to make the edges aware of the consistent and complementary information, this approach inherently overlooks the heterogeneity

of relationships across different views. Consequently, we further consider the following problem

$$\min_{\mathbf{Z}} \frac{1}{2} \sum_v \sum_{i,j} \mu_v \omega_{ij}^v \mathbf{A}_{ij}^v \|\bar{\mathbf{z}}_i - \bar{\mathbf{z}}_j\|_2^2, \qquad (7)$$

where $\mu_v$ is the view-level coefficient which measures the importance of different views, and it is also supposed to be associated with the consistent representation $\mathbf{Z}$.

Then we determine how these coefficients interact with $\mathbf{Z}$. Motivated by the Iterative Reweighted Least Square (IRLS) method [16], we propose to define $\omega_{ij}^v$ as $\omega_{ij}^v = 1/\|\bar{\mathbf{z}}_i - \bar{\mathbf{z}}_j\|_2^\gamma$. In this manner, each edge in every view is reweighted by the consistent representation, which is denoted as $\mathbf{S}_{ij}^v = \mathbf{A}_{ij}^v/\|\bar{\mathbf{z}}_i - \bar{\mathbf{z}}_j\|_2^\gamma$. The problem becomes

$$\min_{\mathbf{Z}, \mathbf{S}^v} \frac{1}{2} \sum_{i,j} \sum_v \mu_v \mathbf{S}_{ij}^v \|\bar{\mathbf{z}}_i - \bar{\mathbf{z}}_j\|_2^2. \qquad (8)$$

As a result, the geometric structure in the consistent embedding space of $\mathbf{Z}$ influences the weighting of edges in each view, and these reweighted edges, in turn, encode structural information into the $\mathbf{Z}$ obtained in the next iteration. For $\mu_v$, we directly set it as a variable, leading to an auto-weighted optimization problem:

$$\min_{\mathbf{Z}, \mathbf{S}^v, \boldsymbol{\mu}} \frac{\lambda_1}{2} \sum_{i,j} \sum_v \mu_v \mathbf{S}_{ij}^v \|\bar{\mathbf{z}}_i - \bar{\mathbf{z}}_j\|_2^2 + \lambda_2 \|\boldsymbol{\mu}\|_2^2,$$
$$s.t. \sum_{v=1}^{V} \mu_v = 1, \mu_v \geq 0, v \in \{1, 2, ..., V\}, \qquad (9)$$

where $\boldsymbol{\mu} = [\mu_1, ..., \mu_V]$, and $\|\boldsymbol{\mu}\|_2^2$ is a regularization term to avoid trivial solutions and promote a balanced representation across views. $\lambda_1$ and $\lambda_2$ are two trade-off parameters. $\mu_v \geq 0$ denotes the weight assigned to the $v$-th view, ensuring that the sum of weights equals 1 to constrain the search space.

Upon establishing the graph smoothing problem, an additional fitting term is required to constrain the similarity between node representations and original features, fully leveraging the feature information. In conclusion, we present a novel graph-smoothing-based framework that integrates node feature learning and graph structure learning in a unified approach:

$$\min_{\mathbf{Z}, \mathbf{S}^v, \boldsymbol{\mu}} \|\mathbf{Z} - \mathbf{X}\|_F^2 + \frac{\lambda_1}{2} \sum_{i,j} \sum_v \mu_v \mathbf{S}_{ij}^v \|\bar{\mathbf{z}}_i - \bar{\mathbf{z}}_j\|_2^2 + \lambda_2 \|\boldsymbol{\mu}\|_2^2,$$
$$s.t. \sum_{v=1}^{V} \mu_v = 1, \mu_v \geq 0, v \in \{1, 2, ..., V\}, \qquad (10)$$

where $\mathbf{S}_{ij}^v = \mathbf{A}_{ij}^v/\|\bar{\mathbf{z}}_i - \bar{\mathbf{z}}_j\|_2^\gamma$.

## 3.2 Cross-view Confluent Message Passing

Jointly optimizing $\mathbf{Z}$, $\mathbf{S}$, and $\boldsymbol{\mu}$ in Problem (10) poses a significant challenge. Unlike conventional optimization problems for GNN model design that solely involve the representation variable, Problem (10) is nonconvex. To obtain an efficient iterative algorithm conducive to back-propagation training, we propose employing an alternating optimization schema to update $\mathbf{Z}$, $\mathbf{S}$, and $\boldsymbol{\mu}$ iteratively.

**Update View-level Coefficients $\boldsymbol{\mu}$:** To update $\boldsymbol{\mu}$, we fix $\mathbf{Z}$ and $\mathbf{S}$, then the objective function in Problem (10) reduces to:

$$\min_{\boldsymbol{\mu}} \mathcal{L} := \sum_v \mu_v t_v + \frac{\lambda_2}{\lambda_1} \|\boldsymbol{\mu}\|_2^2,$$
$$s.t. \sum_{v=1}^{V} \mu_v = 1, \mu_v \geq 0, v \in \{1, 2, ..., V\}, \qquad (11)$$

where $t_v = \frac{1}{2} \sum_{i,j} \mathbf{S}_{ij}^v \|\bar{\mathbf{z}}_i - \bar{\mathbf{z}}_j\|_2^2$. When $\frac{\lambda_2}{\lambda_1} = 0$, the coefficients will be focused on a particular view and it degenerates into the single-view scenario. When $\frac{\lambda_2}{\lambda_1} = +\infty$, it is equivalent to the average fusion strategy, i.e., $\mu_v = \frac{1}{V}$. Otherwise, Problem (11) serves as the problem of minimizing a convex function $f(\cdot)$ on the unit simplex $\triangle = \{\boldsymbol{\mu} \in \mathbb{R}^V : \sum_{v=1}^{V} \mu_v = 1, \boldsymbol{\mu} \geq 0\}$. The Entropic Mirror Descent Algorithm (EMDA) [2] can be used to update $\boldsymbol{\mu}$. For $\boldsymbol{\mu}^{(l)}$ in the $l$-th layer, start with $\boldsymbol{\mu}^1 \in \{\frac{1}{V}, ..., \frac{1}{V}\}$ and generate for $k = 1, ...$ until convergence and obtain the optimal solution as follows:

$$\mu_v^{(l)} \leftarrow \mu_v^{k+1} = \frac{\mu_v^k e^{-R_k f_v'(\boldsymbol{\mu}^k)}}{\sum_{v=1}^{V} \mu_v^k e^{-R_k f_v'(\boldsymbol{\mu}^k)}}, R_k = \frac{\sqrt{2\ln V}}{L_f \sqrt{k}}, \qquad (12)$$

where $f'(\boldsymbol{\mu}) = (f_1'(\boldsymbol{\mu}), ..., f_V'(\boldsymbol{\mu}))^\mathsf{T} \in \partial f(\boldsymbol{\mu})$ and $f_v'(\boldsymbol{\mu}^k) = \frac{2\lambda_2}{\lambda_1} \mu_v^k + t_v$. The objective function $f(\cdot)$ is a convex Lipschitz continuous function with Lipschitz constant $L_f$ with respect to a fixed given norm $\|\cdot\|_1$, i.e., $L_f = \frac{2\lambda_2}{\lambda_1} + \|t\|_1 \geq \|f'(\boldsymbol{\mu})\|_1$, where $t = \{t_1, ..., t_V\}$.

**Update Graph Structure Matrix $\mathbf{S}$:** Similar to IRLS, with fixed $\boldsymbol{\mu}$ and $\mathbf{Z}$, $\mathbf{S}^v$ can be updated by reweighting the adjacency matrix $\mathbf{A}^v$:

$$[\mathbf{S}_{ij}^v]^{(l)} = [\omega_{ij}^v]^{(l)} \mathbf{A}_{ij}^v, \ [\omega_{ij}^v]^{(l)} = 1/\|\bar{\mathbf{z}}_i^{(l)} - \bar{\mathbf{z}}_j^{(l)}\|_2^\gamma. \qquad (13)$$

Through Eq. (13), each edge can dynamically adjust its weight according to the pair-wise representation distance by iteratively updating the matrix $\mathbf{S}^v$.

**Update Node Representation $\mathbf{Z}$:** Finally, $\mathbf{Z}$ can be updated with fixed $\boldsymbol{\mu}$ and $\mathbf{S}$, the subproblem for $\mathbf{Z}$ becomes:

$$\min_{\mathbf{Z}} \mathcal{L} := \|\mathbf{Z} - \mathbf{X}\|_F^2 + \frac{\lambda_1}{2} \sum_{i,j} \sum_v \mu_v \mathbf{S}_{ij}^v \|\bar{\mathbf{z}}_i - \bar{\mathbf{z}}_j\|_2^2. \qquad (14)$$

We calculate the derivative with respect to $\mathbf{Z}$ and set it to zero:

$$\mathbf{Z} = (\mathbf{I} + \lambda_1 \sum_v \mu_v \hat{\mathbf{L}}_\mathbf{S}^v)^{-1}\mathbf{X} = \frac{1}{1+\lambda_1}\left(\mathbf{I} - \frac{\lambda_1}{1+\lambda_1}\sum_v \mu_v \hat{\mathbf{S}}^v\right)^{-1}\mathbf{X}, \quad (15)$$

where $\hat{\mathbf{L}}_\mathbf{S}^v = \mathbf{I} - \hat{\mathbf{S}}^v$, $\hat{\mathbf{S}}^v = (\tilde{\mathbf{D}}^v)^{-\frac{1}{2}}\tilde{\mathbf{S}}^v(\tilde{\mathbf{D}}^v)^{-\frac{1}{2}}$ and $\tilde{\mathbf{S}}^v = \mathbf{S}^v + \mathbf{I}$. Since $\frac{\lambda_1}{1+\lambda_1} < 1$ for $\forall \lambda_1 > 0$, and matrix $\sum_v \mu_v \hat{\mathbf{S}}^v$ has absolute eigenvalues bounded by 1, thus all its positive powers have bounded operator norm. Then the inverse matrix can be decomposed efficiently with $l \to \infty$ using Taylor series, that is $\left(\mathbf{I} - \frac{\lambda_1}{1+\lambda_1}\sum_v \mu_v \hat{\mathbf{S}}^v\right)^{-1} = \lim_{l \to \infty}\sum_{i=0}^{l}\sum_v \left(\frac{\lambda_1}{1+\lambda_1}\right)^i \mu_v [\hat{\mathbf{S}}^v]^{(i)}$. So we consider solving Eq. (15) iteratively and the detailed derivations are in Appendix A. In conclusion, we derive a new message passing mechanism, named Cross-view Confluent Message Passing (CCMP), summarized as follows:

$$\mathbf{z}_i^{(l+1)} = \frac{\lambda_1}{1+\lambda_1}\sum_{i,j}\sum_v \mu_v^{(l)}[\hat{\mathbf{S}}_{ij}^v]^{(l)}\mathbf{z}_j^{(l)} + \frac{1}{1+\lambda_1}\mathbf{x}_i, \qquad (16)$$

where $\mathbf{z}_i^{(l)}$ is the representation of node $i$ from the $l$-th layer.

The CCMP learns the dynamic graph structures $\mathbf{S}$ and the view-level coefficients $\boldsymbol{\mu}$ by extracting information from both consistent representations and relations simultaneously. The graph structures $\mathbf{S}$ adjust the edge weights to evaluate their significance in capturing consistent representations through the connected node pairs' representations. The view-level coefficients $\boldsymbol{\mu}$ associate each edge with various views in a cross-view manner. Therefore, during the message passing process, the overarching consistent representation information and the distinctive information across various views can flow between each edge. Moreover, the term $\frac{1}{1+\lambda_1}\mathbf{x}_i$ in Eq. (16) can be regarded as a residual unit that helps the model mitigate over-smoothing issues.

**Discussion on Robustness:** Due to the graphs' advantages to represent both entities and their relations, constructing graphs on real-world data (e.g. multi-modality data) can lead to better modeling of them. However, there is no such thing as a free lunch, the noisy edges come during the collection or construction process of graphs, which destroys the performance of GNNs. Many studies have pointed out that classical GNNs lack robustness, we then analyze that issue from the perspective of graph smoothing and demonstrate that our proposed method is more robust.

The problem of classical GNN is based on $\ell_2$-norm-based graph smoothing (Eq. (3)), and our method (Eq. (9)) can be rewritten

$$\min_{\mathbf{Z},\mathbf{S}^v,\boldsymbol{\mu}} \frac{\lambda_1}{2}\sum_{i,j}\sum_v \mu_v \mathbf{A}_{ij}^v \|\bar{\mathbf{z}}_i - \bar{\mathbf{z}}_j\|_2^p + \lambda_2 \|\boldsymbol{\mu}\|_2^2,$$
$$s.t. \sum_{v=1}^{V} \mu_v = 1, \mu_v > 0, v \in \{1, 2, ..., V\}, \tag{17}$$

where $p = 2 - \gamma, 1 < p < 2$. In problem (17), we derive a $\ell_{2,p}$-norm-based graph smoothing problem. $\|\bar{\mathbf{z}}_i - \bar{\mathbf{z}}_j\|_2^p$ with $1 < p < 2$ is a more robust regularizer in comaprison to $\|\bar{\mathbf{z}}_i - \bar{\mathbf{z}}_j\|_2^2$. Particularly, a smaller $p$ leads to a shaper regularizer, and when $p \to 1$, $\|\bar{\mathbf{z}}_i - \bar{\mathbf{z}}_j\|_2^p$ approaches a discrete operator: if $\bar{\mathbf{z}}_i - \bar{\mathbf{z}}_j \neq \mathbf{0}$. Therefore, $\sum_{i,j}\mathbf{A}_{ij}\|\bar{\mathbf{z}}_i - \bar{\mathbf{z}}_j\|_2^p$ is an effective relaxation of $\sum_{i,j}\mathbf{A}_{ij}\mathbb{1}\{\bar{\mathbf{z}}_i - \bar{\mathbf{z}}_j \neq \mathbf{0}\}$ when $1 < p < 2$. Additionally, minimizing $\sum_{i,j}\mathbf{A}_{ij}\|\bar{\mathbf{z}}_i - \bar{\mathbf{z}}_j\|_2^p$ is able to isolate the outliers in $\{\bar{\mathbf{z}}_i - \bar{\mathbf{z}}_j\}_{i,j=1}^n$.

Specifically, Problem (17) can be regarded as weighting $\mathbf{A}_{ij}^v$ with $1/\|\bar{\mathbf{z}}_i - \bar{\mathbf{z}}_j\|_2^{2-p}$ based on the $\ell_2$-norm-based graph smoothing. In this way, the variation of node embeddings on edges (measured by the norm of graph gradient) can be used to detect the existence of outlier edges according to homophily assumption [1, 10]. When nodes $i$ and $j$ are connected but dissimilar, the resulting edge is likely an outlier edge. Consequently, a small weight would be assigned to this edge accordingly, meaning that $\mathbf{A}_{ij}/\|\bar{\mathbf{z}}_i - \bar{\mathbf{z}}_j\|_2^{2-p}$ would have a small value. Thus the robustness of the model is greatly enhanced by reducing the impact of outlier edges.

## 3.3 The Overall Model Architecture

In this subsection, we design the architecture of CGNN using Cross-view Confulent Message Passing. Given node features $\mathbf{x}_i \in \mathbb{R}^{n \times m}$ of node $i$, the maximum layer number of $L$, and view-level coefficients update function $\mathrm{U}(\cdot)$ given by Eq. (12), we perform the proposed

architecture:

$$\mathbf{z}_i^{(0)} = \mathrm{ReLU}(\mathbf{x}_i \mathbf{W}_1),$$
$$\mu_v^{(l)} = \mathrm{U}(\mu_v^1, \boldsymbol{t}^{(l)}, \lambda_1, \lambda_2),$$
$$[\omega_{ij}^v]^{(l)} = 1/\|\bar{\mathbf{z}}_i^{(l)} - \bar{\mathbf{z}}_j^{(l)}\|_2^{\gamma},$$
$$[\mathbf{S}_{ij}^v]^{(l)} = [\omega_{ij}^v]^{(l)}\mathbf{A}_{ij}^v, \tag{18}$$
$$\mathbf{z}_i^{(l+1)} = \frac{\lambda_1}{1+\lambda_1}\sum_{i,j}\sum_v \mu_v^{(l)}[\hat{\mathbf{S}}_{ij}^v]^{(l)}\mathbf{z}_j^{(l)} + \frac{1}{1+\lambda_1}\mathbf{z}_i^{(0)},$$
$$\mathbf{z}_{out} = \mathrm{softmax}(\mathbf{z}_i^{(L)}\mathbf{W}_2),$$

where $l = 0, 1, ..., L-1$ and $t_v^{(l)} = \frac{1}{2}\sum_{i,j}[\mathbf{S}_{ij}^v]^{(l)}\|\bar{\mathbf{z}}_i^{(l)} - \bar{\mathbf{z}}_j^{(l)}\|_2^2$, $\boldsymbol{t}^{(l)} = \{t_1^{(l)}, ..., t_V^{(l)}\}$. $\mathbf{z}_{out}$ is the output representation and $c$ denotes the number of classes. $\mathbf{W}_1 \in \mathbb{R}^{m \times d}$ and $\mathbf{W}_2 \in \mathbb{R}^{d \times c}$ are the trainable weight of the neural network. The loss function is chosen as the cross-entropy loss defined by the $\mathbf{z}_{out}$ and labels for training data.

**Computation Complexity:** Recall that the number of nodes is $n$, node feature dimension is $m$, and $|\mathcal{E}|$ is the number of edges. The time complexity of transformation for input is $O(nmd)$, and that of the one for output is $O(ndc)$. The time complexity of coefficient $\boldsymbol{\mu}$ is $O(KV|\mathcal{E}|d)$, where $K$ is the iteration number. For $\mathbf{S}$ and $\mathbf{Z}$, it has a time complexity of $O(LV(KV|\mathcal{E}|d + |\mathcal{E}|d))$, so the overall time complexity is $O((nm + LKV^2|\mathcal{E}|)d)$ when $c \ll m$. Generally, our method is efficient.

## 4 EXPERIMENTS

In this section, we conduct experiments to evaluate CGNN by answering the following Research Questions (RQ):

- **RQ1:** Does CGNN outperform competitors in semi-supervised classification when processing various graph data?
- **RQ2:** How do reweighted edges boost the robustness?
- **RQ3:** How does the view-level dynamic coefficients work?
- **RQ4:** How does each component or hyperparameter affect the performance of CGNN?

## 4.1 Experimental Settings

*4.1.1 Datasets.* To validate the effectiveness of the proposed model on various data types, we conduct experiments on six single-view datasets (Cora, Citeseer, Pubmed, ACM, BlogCatalog, UAI); ten multi-view ones including four multi-relational (ACM, DBLP, IMDB, YELP), three multi-attribute (MINST, HW, Animals) and three multi-modality (BDGP, ESP-Game, MIRFlickr). To distinguish two different ACM datasets, we use ACM-S to denote ACM for single-view and ACM-M to denote ACM for multi-relational.

*4.1.2 Compared Methods.* For verifying the superiority of the proposed model, we compare CGNN with nine GNN-based models used for single-view graph (GCN [22], GAT [37], SGC [45], APPNP [13], ScGCN [29], AdaGCN [34], AMGCN [43], SSGC [57], DefGCN [32]); seven approaches designed for multi-relational graphs (HAN [42], DMGI [31], IGNN [15], MRGCN [18], SSDCM [30], MHGCN [54], AMOGCN [8]); eight methods devised for multi-attribute and multi-modality graphs (MVAR [36], Co-GCN [25], HLR-M$^2$VS [48], ERL-MVSC [17], DSRL [41], LGCN-FF [7], IMvGCN [46], JFGCN [6]).

**Table 1: ACC (mean and std%) of ten models on single-view graph, where the best and the second-best performance are highlighted in bold and underlined, respectively. OOM is Out-of-Memory (24GB).**

| Methods/Datasets | GCN | GAT | SGC | APPNP | ScGCN | AdaGCN | AMGCN | SSGC | DefGCN | CGNN (Ours) |
|---|---|---|---|---|---|---|---|---|---|---|
| Cora | 79.32 (0.91) | 79.11 (0.83) | 77.14 (0.05) | 77.94 (0.13) | 77.86 (0.92) | 75.42 (0.03) | 79.32 (0.67) | 80.24 (0.81) | 77.83 (1.03) | **81.50 (0.22)** |
| Citeseer | 69.25 (0.54) | 68.37 (0.53) | 67.08 (0.01) | 66.83 (0.03) | 67.91 (0.25) | 66.78 (0.25) | **71.92 (0.81)** | 71.03 (0.63) | 67.59 (1.75) | 69.60 (1.07) |
| Pubmed | 77.65 (2.37) | 77.62 (2.21) | 78.27 (0.89) | 78.55 (1.13) | 76.70 (0.52) | 77.54 (0.48) | OOM | 78.93 (0.36) | 73.92 (0.52) | **80.13 (0.54)** |
| ACM-S | 88.52 (0.73) | 84.63 (0.54) | 80.42 (0.13) | 83.26 (0.13) | 87.52 (0.71) | 85.10 (0.96) | 89.93 (0.42) | 85.42 (0.28) | 87.83 (0.30) | **90.23 (0.21)** |
| BlogCatalog | 84.67 (1.14) | 65.30 (1.72) | 73.52 (0.28) | 81.75 (0.10) | 68.58 (1.43) | 80.71 (0.79) | 85.86 (0.90) | 82.30 (0.84) | 82.84 (3.47) | **96.17 (0.19)** |
| UAI | 53.67 (2.12) | 49.71 (3.02) | 56.53 (3.55) | 60.23 (0.16) | 37.72 (3.68) | 45.92 (7.61) | 64.32 (0.95) | 59.73 (1.04) | 57.82 (1.84) | **67.40 (0.85)** |

**Table 2: Macro-F1 and Micro-F1 (mean and std%) of nine networks with various percentages of training samples on multi-relational graphs, in which the best and the second-best performance are highlighted in bold and underlined, respectively.**

| Datasets | Metrics | Training | GCN | HAN | DMGI | IGNN | MRGCN | SSDCM | MHGCN | AMOGCN | CGNN |
|---|---|---|---|---|---|---|---|---|---|---|---|
| ACM-M | Macro-F1 | 20% | 76.66 (5.19) | 87.95 (0.42) | 66.73 (1.84) | 82.90 (0.03) | 87.58 (0.19) | 84.34 (3.46) | 60.15 (1.50) | 90.14 (0.48) | **93.84 (0.30)** |
| | | 40% | 78.34 (3.76) | 91.28 (0.33) | 71.17 (2.24) | 85.01 (1.01) | 88.44 (0.20) | 85.05 (3.66) | 60.72 (0.40) | 91.03 (0.50) | **94.52 (0.37)** |
| | | 60% | 79.10 (3.58) | 89.22 (0.53) | 68.38 (1.57) | 87.29 (0.07) | 91.49 (0.43) | 86.44 (1.37) | 90.90 (2.78) | 90.96 (0.78) | **94.83 (0.29)** |
| | Micro-F1 | 20% | 78.01 (4.54) | 87.98 (0.38) | 70.43 (1.13) | 82.70 (0.03) | 87.45 (0.24) | 85.23 (2.86) | 72.80 (0.65) | 90.01 (0.50) | **93.59 (0.29)** |
| | | 40% | 79.36 (3.27) | 91.20 (0.33) | 74.06 (1.50) | 84.91 (1.01) | 88.32 (0.20) | 85.77 (3.09) | 73.17 (1.05) | 90.97 (0.51) | **94.48 (0.37)** |
| | | 60% | 80.01 (3.23) | 89.21 (0.54) | 71.80 (1.01) | 87.39 (0.07) | 91.43 (0.44) | 86.96 (1.04) | 90.86 (2.65) | 90.81 (0.79) | **94.77 (0.29)** |
| DBLP | Macro-F1 | 20% | 90.70 (0.64) | 89.30 (0.21) | 75.35 (1.28) | 86.81 (0.01) | 89.49 (1.61) | 55.72 (2.71) | 92.52 (0.29) | 92.27 (0.42) | **93.01 (0.12)** |
| | | 40% | 89.86 (0.26) | 90.02 (0.34) | 81.47 (0.76) | 88.40 (0.01) | 91.15 (0.08) | 79.88 (2.13) | 92.00 (0.22) | 92.24 (0.29) | **92.71 (0.15)** |
| | | 60% | 90.26 (0.50) | 90.70 (0.27) | 77.72 (1.41) | 87.76 (0.00) | 91.07 (0.13) | 80.69 (2.05) | 92.02 (0.48) | 92.36 (0.18) | **92.45 (0.28)** |
| | Micro-F1 | 20% | 91.38 (0.52) | 90.44 (0.20) | 81.21 (0.72) | 87.50 (0.01) | 90.47 (1.23) | 62.82 (3.90) | 92.97 (0.23) | 92.80 (0.37) | **93.19 (0.12)** |
| | | 40% | 90.62 (0.18) | 90.46 (0.28) | 83.77 (0.51) | 88.41 (0.01) | 91.71 (0.12) | 80.64 (2.11) | 92.74 (0.25) | 92.70 (0.26) | **93.10 (0.15)** |
| | | 60% | 90.97 (0.56) | 91.20 (0.29) | 82.81 (0.78) | 88.33 (0.00) | 91.59 (0.12) | 81.37 (2.03) | 91.28 (0.45) | 92.73 (0.16) | **92.83 (0.27)** |
| IMDB | Macro-F1 | 20% | 23.57 (0.04) | 23.99 (1.19) | 38.28 (3.12) | 45.31 (0.00) | 45.17 (2.31) | 35.29 (3.54) | 51.38 (1.24) | 49.11 (0.75) | **52.74 (2.59)** |
| | | 40% | 24.38 (0.53) | 23.11 (1.91) | 50.81 (0.01) | 50.80 (0.10) | 45.74 (1.28) | 38.86 (0.02) | 52.04 (0.26) | 50.86 (0.74) | **52.47 (0.68)** |
| | | 60% | 25.49 (1.36) | 24.90 (1.62) | 40.37 (1.90) | 53.60 (0.01) | 49.15 (2.63) | 39.44 (0.03) | **52.44 (1.53)** | 52.38 (0.44) | 50.79 (1.55) |
| | Micro-F1 | 20% | 54.60 (0.01) | 55.90 (1.21) | 57.03 (0.42) | 54.81 (0.00) | 47.69 (2.28) | 50.83 (2.69) | 62.12 (0.95) | 61.03(1.27) | **62.96 (0.38)** |
| | | 40% | 54.72 (0.02) | 54.25 (0.77) | 59.22 (0.01) | 59.18 (0.03) | 48.27 (0.90) | 51.56 (0.01) | 61.44 (0.36) | 63.81 (0.20) | **64.61 (0.32)** |
| | | 60% | 54.94 (0.27) | 56.21 (0.57) | 58.86 (0.28) | 61.22 (0.11) | 51.81 (3.51) | 54.04 (0.02) | 62.67 (0.46) | 62.08 (0.26) | **63.52 (0.70)** |
| YELP | Macro-F1 | 20% | 55.46 (0.89) | 55.39 (4.52) | 52.72 (2.27) | 71.40 (0.01) | 54.35 (0.39) | 55.86 (2.99) | 60.85 (1.02) | 70.77 (2.32) | **93.18 (0.18)** |
| | | 40% | 55.65 (1.01) | 55.59 (4.80) | 55.54 (3.24) | 73.33 (0.01) | 54.74 (0.91) | 69.54 (2.04) | 60.07 (1.01) | 70.97 (1.81) | **93.50 (0.32)** |
| | | 60% | 60.44 (2.17) | 56.26 (5.77) | 53.94 (3.25) | 75.30 (0.01) | 53.54 (0.04) | 69.44 (2.06) | 56.62 (1.18) | 73.26 (1.08) | **93.75 (0.47)** |
| | Micro-F1 | 20% | 74.02 (0.35) | 68.00 (5.03) | 69.52 (0.68) | 75.01 (0.01) | 73.70 (0.46) | 68.87 (5.54) | 73.28 (0.24) | 77.43 (0.36) | **92.40 (0.23)** |
| | | 40% | 74.13 (0.49) | 69.69 (6.25) | 72.60 (0.25) | 75.91 (0.01) | 73.53 (0.50) | 75.77 (2.10) | 73.01 (0.49) | 78.81 (0.19) | **92.67 (0.34)** |
| | | 60% | 75.70 (0.94) | 68.02 (6.63) | 71.00 (0.39) | 77.51 (0.01) | 72.55 (0.06) | 74.91 (2.19) | 73.21 (1.00) | 79.69 (0.47) | **93.07 (0.42)** |

*4.1.3 Experimental Settings.* For the performance evaluation, we conduct 5 runs for semi-supervised classification on all datasets and record the mean and standard deviation. The detailed experimental settings are provided in Appendix B.

## 4.2 Performance on Various Graph data (RQ1)

In the subsection, we focus on constructing semi-supervised classification experiments to validate whether the proposed CGNN can effectively process various types of graph data.

*4.2.1 Comparison on Single-view Graph.* When the number of views equals 1, the formula (10) can be utilized for the datasets with only one view. For ten methods devised for single-view graph, 20 labeled samples per class are randomly chosen for training, 500 samples are selected for validation, and 1000 samples are used for testing. Comparative results are shown in Table 1, and we summarize the following several observations from this table:

- Our model CGNN outperforms other competitors and achieves the optimal performance on most datasets.
- Especially on BlogCatalog, we improve the accuracy by 10.31% over the second-best method AMGCN.

It is noted that compared with other models with graph enhancement GAT, ScGCN and DefGCN, CGNN obtains significant improvement. These results can be attributed to the effectiveness of CGNN with reweighted edges guided by the final representation.

*4.2.2 Comparison on Multi-view Graphs.* For the sake of displaying the advancement of our method in handling multi-view graphs, we conduct diverse experiments on three kinds of datasets with multiple views. Experimental setups and analyses on these datasets are as follows: 1) **Multi-relational Graphs** denote that there are multiple relationship types between nodes in the graph structure. We record Macro-F1 and Miscro-F1 of all models in Table 2. Here, the training percentage varies in {20%, 40%, 60%}, and the validation and test ratios are fixed at 10% and the rest, respectively. From this

**Table 3: ACC (mean and std%) of nine approaches on multi-attribute and multi-modality graphs, in which the optimal and the suboptimal performance are highlighted in bold and underlined, respectively.**

| Methods/Datasets | MVAR | Co-GCN | HLR-M$^2$VS | ERL-MVSC | DSRL | LGCN-FF | IMvGCN | JFGCN | CGNN |
|---|---|---|---|---|---|---|---|---|---|
| MNIST | 85.54 (1.93) | 90.41 (1.47) | 83.06 (2.53) | 91.36 (0.66) | 89.04 (6.48) | 89.96 (0.34) | 91.92 (1.45) | 89.33 (1.89) | **92.65 (0.30)** |
| HW | 78.43 (2.16) | 91.44 (4.39) | 86.04 (2.00) | 89.81 (1.08) | 95.55 (2.43) | 96.23 (0.43) | 92.53 (1.28) | 92.98 (4.57) | **97.50 (0.06)** |
| Animals | 81.51 (0.54) | 79.72 (1.38) | 72.87 (0.48) | 69.98 (0.57) | 80.19 (4.34) | 74.42 (1.02) | 68.81 (0.27) | 80.31 (0.17) | **84.04 (0.04)** |
| BDGP | 94.81 (1.46) | 94.56 (1.73) | 94.31 (1.18) | 93.48 (0.81) | 98.58 (0.91) | 98.74 (0.16) | 93.34 (0.45) | 98.52 (0.48) | **99.15 (0.05)** |
| ESP-Game | 79.15 (2.62) | 75.94 (3.51) | 66.97 (0.67) | 68.56 (0.42) | 79.85 (6.01) | 68.80 (0.37) | 71.34 (0.74) | 53.96 (4.39) | **82.03 (0.38)** |
| MIRFlickr | 67.15 (0.49) | 59.24 (2.69) | 57.01 (0.74) | 58.93 (0.62) | 68.71 (7.62) | 41.24 (0.74) | 58.89 (1.02) | 48.36 (3.54) | **70.02 (0.28)** |

table, we can observe that for most methods, performance improves with an increase in the number of training samples. Among them, CGNN gains the best Macro-F1 and Miscro-F1 on most datasets for distinct ratios of training samples; 2) **Multi-attribute and Multi-modality Graphs** are constructed based on various data features, where each feature corresponds to a graph. We run nine approaches with the fixed training/validation/test split as 10%/10%/80%, and experimental results are presented in Table 3. It is obvious that the champion and runners-up are typically GNN-based methods, validating the role of message passing in performance improvement. Among them, CGNN is the best due to its advanced cross-view confluent message passing mechanism.

### 4.3 Robustness Analysis of Graph Structures (RQ2)

To verify the robustness discussed in Section 3.2, we use the state-of-the-art adversarial attacks, Mettack [59], to perturb the graph data with a 25% perturbation rate. Specific implementation details and robustness analysis on multi-view datasets are available in Appendix C. The adversary usually adds adversarial edges rather than deleting edges in structural attacks [59]. We design a visualization task using the edge-level coefficients $\omega$ in Eq. (18) to calculate the edge weights. For a fair comparison, we compute the weight for each edge in the last layer and normalize them. We visualize the weight density distribution of normal and adversarial edges with and without CCMP as well as from two datasets respectively in Figure 3. From the figure, it is evident that the peak of the density for adversarial edges shifts towards lower weights, whereas the peak for normal edges remains relatively constant. Meanwhile, the classification accuracy reduction after the attack has been noticeably alleviated. Consistent with earlier discussion, the outlier edge is allocated a smaller weight, alleviating its impact and validating the robustness of the learned graph structure.

### 4.4 View-level Coefficient Visualization (RQ3)

In this subsection, we explore whether the proposed view-level coefficients generated by CGNN are reasonable. For datasets with more than two views, we assess the classification performance for each individual view and rank them in descending order, starting from the highest F1 score as the 1st F1, followed by the 2nd F1, and so on up to the 6th F1. The coefficient $\mu_v$ of each view (denoted by color length) and its corresponding F1 rank are presented in Figure 4. We can observe that obtaining better performance is basically associated with larger coefficient values. Beyond that, we calculated Pearson correlation coefficients (PCCs) between the F1 scores and

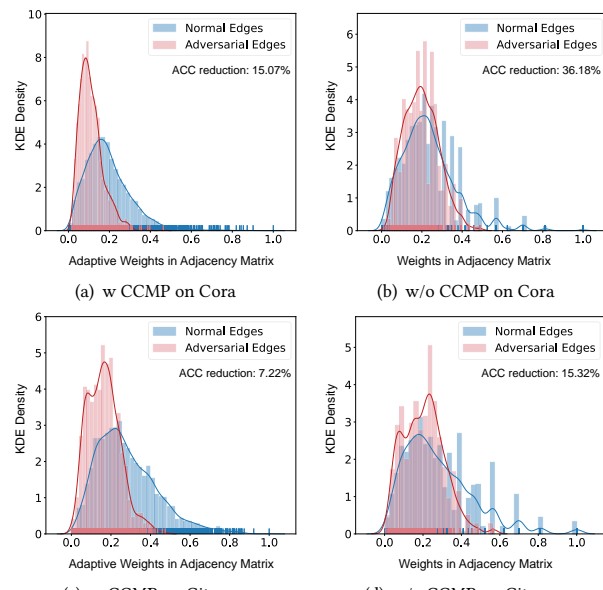

**Figure 3: Weight density distributions of normal and adversarial edges on the learned graph. We implement CGNN w and w/o CCMP on Cora and Citeseer datasets.**

**Figure 4: View-level Coefficient $\mu_v$ of each view. Different colors indicate the ranking of Macro-F1 for each single view, e.g., the 1st F1 corresponds to the highest Macro-F1.**

coefficient $\mu$ for each dataset. The positive correlations observed in all datasets for the PCCs suggest that the learned view-level coefficients effectively capture the importance of different views.

### 4.5 Component and Parameter Analysis (RQ4)

*4.5.1 Ablation Study.* To evaluate the effectiveness of the proposed CCMP, we compare CGNN with one variant in single graph scenario: BaseGNN (with message passing of GCN) and three variants in multi-view scenario: BaseGNN, CGNN-S (without considering

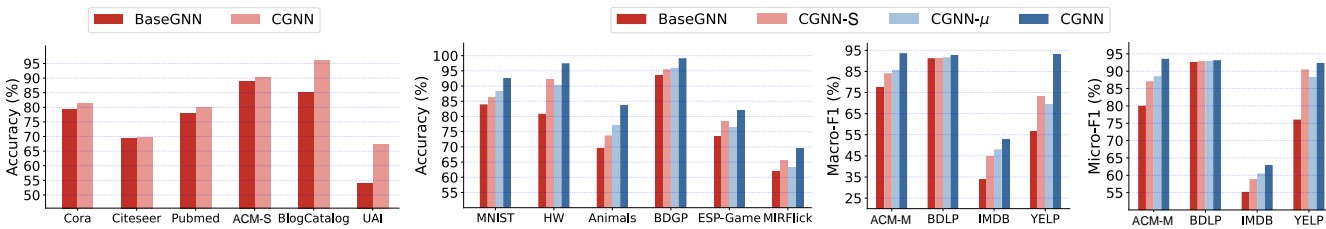

Figure 5: Performance comparison of CGNN and its variants on various datasets.

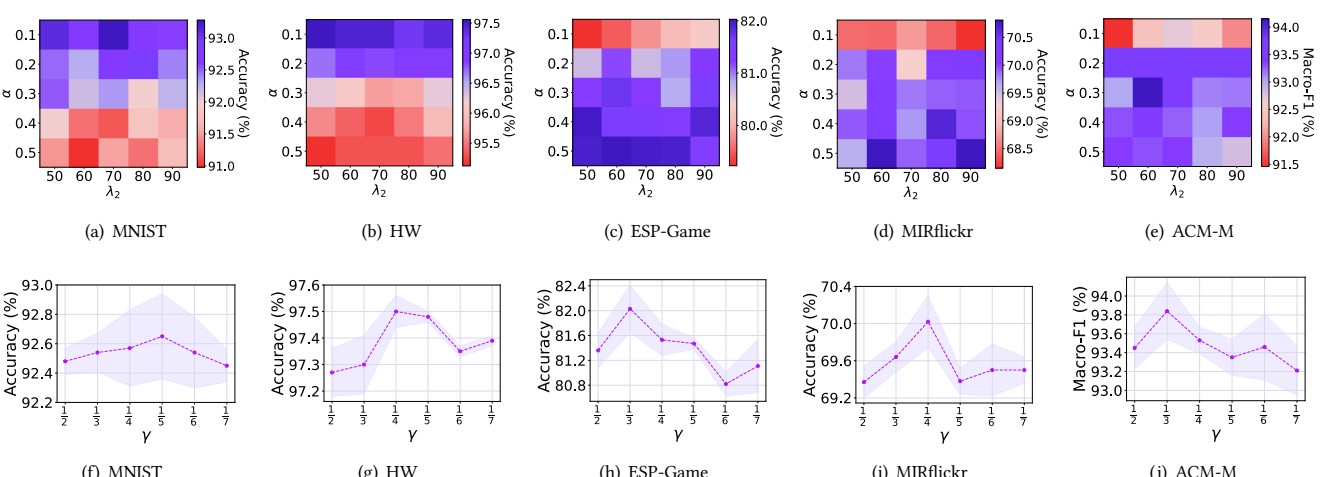

(a) MNIST  (b) HW  (c) ESP-Game  (d) MIRflickr  (e) ACM-M

(f) MNIST  (g) HW  (h) ESP-Game  (i) MIRflickr  (j) ACM-M

Figure 6: Parameter sensitivity on five multi-view graph datasets. (a)-(e) Performance with different combinations of $\alpha$ and $\lambda_2$. (f)-(j) Performance curves as $\gamma$ ranges in $\{\frac{1}{2}, \cdots, \frac{1}{7}\}$.

dynamic graph structures in CCMP) and CGNN-$\mu$ (without considering view-level coefficients in CCMP). Figure 5 presents the node classification performance across all datasets, which demonstrates:

- For the single graph scenario, the adoption of CCMP leads to performance improvements on all datasets. It indicates that the learned graph structures via edge-level coefficients enable a more effective capture of structural information for each view.

- For the multi-view graph scenario, CGNN outperforms each variant in terms of classification performance, validating that the proposed CCMP can efficiently and rationally leverage consistency and complementarity information across views.

4.5.2 *Parameter Sensitivity.* In this subsection, we conduct the parameter sensitivity to analyze the impact of these parameters on the models. Due to page limitations, we present results on 2 multi-attribute datasets, 2 multi-modality datasets, and 1 multi-relational dataset. Results for the remaining datasets can be found in Appendix C. This paper contains three main hyperparameters: $\alpha = \frac{1}{1+\lambda_1}$ balancing the multi-view graph smoothing term, $\lambda_2$ adjusting the regularization term $\|\boldsymbol{\mu}\|_2^2$, and $\gamma$ controlling the norm of edge-level coefficients $\boldsymbol{\omega}$. Figure 6 (a)-(e) plot the performance of CGNN w.r.t. $(\alpha, \lambda_2)$ with the fixed $\gamma$. It is obvious that the model's peak is achieved when $\alpha = 0.2$ for the multi-relational and multi-modality datasets,

and the optimal value occurs when $\alpha$ varies from 0.1 to 0.2 for the multi-attribute datasets. This indicates that the appropriate $\alpha$ can promote the performance and validate the effectiveness of the graph smoothing term. For the parameter $\lambda_2$, the model performance varies as it is changed, implying the role of this term balancing multiple views. Figure 6 (f)-(j) presents the effect of $\gamma$ on the performance. The model stays at a promising level with slight fluctuations as $\gamma$ varies, which supports the validity of the dynamic graph structures driven by edge-level coefficients.

## 5 CONCLUSION

In this paper, we revealed the key to designing multi-view message passing and connecting it with the graph smoothing problem of GNNs. Motivated by our findings, we introduced a novel optimization objective that fully utilized the interaction between graph structures and consistent representation. By iteratively optimizing the objective, we naturally derived a Cross-view Confluent Message Passing (CCMP), seamlessly integrating it into the Confluent Graph Neural Networks (CGNN). The dynamic weight facilitated the flow of consistency and complementarity information across views along edges during the aggregation process. Extensive experiments on node classification tasks on various datasets demonstrated the effectiveness of the proposed CGNN, and also exhibited the robustness on graphs with noisy edges.

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
