# OpenReview forum: "Enhancing Multi-view Graph Neural Network with Cross-view Confluent Message Passing"
_acmmm.org/ACMMM/2024/Conference — MM2024 Poster_

### Official Review · Reviewer_XLF9 · 2024-05-21

**Rating:** 5
**Confidence:** 4

**Summary:**

The paper studies the inherent problem of multi-view message passing. The authors analyzed the graph smoothing problem of multi-view GNN cannot fully utilize the interaction between consistent representation and relations. The solution is to redesign the optimization problem that enable message passing at both the edge and view levels. By optimizing the proposed objective, the new message passing mechanism is obtained. Extensive experiments are employed to demonstrate the effectiveness of CGCN. Particularly, CGCN achieves robustness in scenarios with high attack rates while maintaining competitive performance on clean graphs of different data types.

**Strengths:**

S1. The paper is generally of high quality.
S2. This paper naturally proposes a GNN architecture specifically designed for multi-graph data, inspired by the graph smoothing problem.
S3. The concept of 'confluent message passing' seems reliable and novel, different from existing pre- or post- fusion-based frameworks.
S3. The robustness is validated through experiments against adversarial attack, which is persuasive.

**Limitations:**

L1. Some experimental details should be reported. For example, it seems that not all adopted datasets are graph-structured, the process of graph construction needs to be further explained.
L2. Some important results, such as the adversarial attack experiments on multi-view graphs, are placed in the Appendix rather than in the main paper. The authors are advised to reorganize the paper to show these results in the main body.
L3. This is just a suggestion. Datasets like Cora only contain single graphs. The authors could try constructing multiple graphs using some graph augmentation techniques. I am curious to see how the CGNN would perform.

**Suitability:**

3

---

### Official Review · Reviewer_3yWP · 2024-05-22

**Rating:** 5
**Confidence:** 3

**Summary:**

The authors propose a novel GNN model that establishes a node-level message passing by introducing an enhanced multi-view graph smoothing problem. The motivation is to connect multi-view message passing with the graph smoothing problem of GNNs, aiming to address the shortcomings of traditional multi-view GNNs from an optimization perspective. Extensive experiments demonstrate the proposed model’s effectiveness and robustness.

**Strengths:**

1.	The paper is structured clearly and the writing is coherent.
2.	The study presents a novel approach by extending classical message passing to multi-view scenarios, solving limitations of multi-view GNNs.
3.	The proposed CGNN improves performance and robustness from both node-level and view-level aspects.
4.	The experimental section is comprehensive, comparing the proposed method against many SOTA competitors across various datasets, including large-scale ones.
5.	The code has been made available for reproducibility and is well-organized according to the authors.

**Limitations:**

There are some questions for the authors:
1. I wonder if the proposed CCMP can be connected to other message-passing mechanisms.
3. Although the paper boosts the performance in various semi-supervised classification tasks, can it also achieve excellent results on other tasks, such as link prediction?
3. On some datasets, the improvement of CGNN compared with other models is not obvious (such as IMDB and Citeseer) and on the YELP dataset, the model obviously outperforms the suboptimal one. The reasons should be clarified.

**Suitability:**

3

---

### Official Review · Reviewer_7Pkv · 2024-05-24

**Rating:** 3
**Confidence:** 3

**Summary:**

In response to the insufficient consideration of the interaction between views in existing multi-view graphs, the paper proposes a multi-view GNN framework, which understands the mechanism of graph message propagation from the perspective of graph smoothing and considers the mutually reinforcing interaction between multi-view graph structures and consistent feature representations. Numerous experiments have verified its effectiveness and its robustness to noise immunity.

**Strengths:**

1. From the perspective of novelty, considering message passing from the perspective of graph smoothing is interesting.

2. Open source code helps to enhance the understanding of the paper.

3. Thorough experiments and a clear algorithmic process.

**Limitations:**

1. The lack of citation for previous work at line 300 seems to be a crucial omission.

2. Eq.(2) can also be referred to as Laplacian Eigenmaps (LE), or a key element of spectral clustering. Is this related to graph smoothing?

3. Eq.(9) seems to share some similarities with the ideas presented in the following paper. The authors need to provide further explanation.

    AAAI17: Multi-View Clustering and Semi-Supervised Classification with Adaptive Neighbours.

4. Combining the idea of iteratively solving variables from shallow methods with the automatic differentiation concept of deep methods is a good attempt. However, can the authors provide a deeper insight about GNN, perhaps purely from the perspective of shallow or deep methods?

5. Perhaps it would be better to add references to contrasting algorithms and datasets in Table.

**Suitability:**

3

---

### Official Review · Reviewer_aZY9 · 2024-05-24

**Rating:** 4
**Confidence:** 3

**Summary:**

The paper proposes a novel multi-view Graph Neural Network (GNN) framework called Confluent Graph Neural Networks (CGNN) with Cross-view Confluent Message Passing (CCMP). The key contributions are:
- 1) revealing the key to designing effective multi-view message passing from a graph smoothing perspective;
- 2) introducing an explicit optimization objective that considers the interaction between multi-view graph structures and consistent representations;
- 3) deriving the CCMP layers through alternating optimization of the proposed objective, which includes three sub-modules (S-Block, $\mu$-Block, and Z-Block) that enable interactions between graph structures and consistent representations.
Extensive experiments demonstrate the superior effectiveness and robustness of the CGNN framework compared to existing multi-view GNN methods.

**Strengths:**

Strengths:

- The motivation is clear, and paper writing is excellent and easy to follow;
- The proposed method, Confluent Graph Neural Networks (CGNN) with Cross-view Confluent Message Passing (CCMP), can effectively integrate graph structures and consistent representations across multiple views;
- Additionally, the CCMP learns dynamic graph structures and view-level coefficients by simultaneously extracting information from consistent representations and relations, enabling consistency and complementarity information across views during the aggregation process;
- Robustness analysis and computation complexity are provided. Specifically, the experimental results demonstrate that the proposed method can effectively mitigate the impact of noisy edges by allocating smaller weights to outlier edges and verify the hypothesis on adversarial attack experiments;

**Limitations:**

Weakness:
- In the single-view graph experiments, more effective and novel GNN methods, such as GraphCon[1], GREAD[2], and ACM-GCN, should be considered for fair comparisons. This question must also be considered in the other two parts of the experiments.
- The computational complexity of CGNN may be overhead compared to other more straightforward graph neural network models, where incorporating multiple sub-modules and the interaction between graph structures and consistent representations in the proposed Cross-view Confluent Message Passing mechanism leads to increased training time and resource requirements.
- The performance of CGNN on extremely large or high-dimensional datasets needs to be thoroughly evaluated, raising questions about its scalability to real-world applications with massive amounts of data.
- Some small grammar issues should be solved:
   - The second approaches -> approach
   - The third modeling algorithms construct -> The third modeling algorithm constructs
   - from two datasets respectively -> from two datasets, respectively,

**Suitability:**

3

---

### Meta-Review · Area_Chair_DRJB · 2024-06-27

**Recommendation:** Accept (Poster)
**Confidence:** 4

**Metareview:**

According to all the review comments, rebuttals, discussions and final ratings, the majority of the reviewers gave positive ratings to this paper and the concerns were well addressed. I am happy to recommend to accept this paper. Please carefully revise the final manuscript according to the comments and discussions.